# The History of Anti-Trypanosome Vaccine Development Shows That Highly Immunogenic and Exposed Pathogen-Derived Antigens Are Not Necessarily Good Target Candidates: Enolase and ISG75 as Examples

**DOI:** 10.3390/pathogens10081050

**Published:** 2021-08-19

**Authors:** Stefan Magez, Zeng Li, Hang Thi Thu Nguyen, Joar Esteban Pinto Torres, Pieter Van Wielendaele, Magdalena Radwanska, Jakub Began, Sebastian Zoll, Yann G.-J. Sterckx

**Affiliations:** 1Laboratory of Cellular and Molecular Immunology, Department of Bioengineering Sciences, Vrije Universiteit Brussel, Pleinlaan 2, 1050 Brussels, Belgium; zeng.li@vub.be (Z.L.); hang.nguyenthithu@ugent.be (H.T.T.N.); Joar.Pinto@vub.be (J.E.P.T.); 2Department of Biochemistry and Microbiology, Ghent University, Ledeganckstraat 35, 9000 Ghent, Belgium; 3Laboratory for Biomedical Research, Department of Molecular Biotechnology, Environment Technology and Food Technology, Ghent University Global Campus, Songdomunhwa-Ro 119-5, Yeonsu-Gu, Incheon 406-840, Korea; Magdalena.Radwanska@ghent.ac.kr; 4Laboratory of Medical Biochemistry (LMB) and the Infla-Med Centre of Excellence, University of Antwerp, Campus Drie Eiken, Universiteitsplein 1, 2610 Wilrijk, Belgium; Pieter.VanWielendaele@uantwerpen.be (P.V.W.); yann.sterckx@uantwerpen.be (Y.G.-J.S.); 5Department of Biomedical Molecular Biology, Ghent University, Technologiepark Zwijnaarde 71, 9000 Ghent, Belgium; 6Laboratory of Structural Parasitology, Institute of Organic Chemistry and Biochemistry, Academy of Sciences of the Czech Republic, Flemingovo Namesti 2, 16610 Prague 6, Czech Republic; Jakub.Began@unil.ch (J.B.); sebastian.zoll@uochb.cas.cz (S.Z.)

**Keywords:** trypanosomosis, vaccination, enolase, ISG75

## Abstract

Salivarian trypanosomes comprise a group of extracellular anthroponotic and zoonotic parasites. The only sustainable method for global control of these infection is through vaccination of livestock animals. Despite multiple reports describing promising laboratory results, no single field-applicable solution has been successful so far. Conventionally, vaccine research focusses mostly on exposed immunogenic antigens, or the structural molecular knowledge of surface exposed invariant immunogens. Unfortunately, extracellular parasites (or parasites with extracellular life stages) have devised efficient defense systems against host antibody attacks, so they can deal with the mammalian humoral immune response. In the case of trypanosomes, it appears that these mechanisms have been perfected, leading to vaccine failure in natural hosts. Here, we provide two examples of potential vaccine candidates that, despite being immunogenic and accessible to the immune system, failed to induce a functionally protective memory response. First, trypanosomal enolase was tested as a vaccine candidate, as it was recently characterized as a highly conserved enzyme that is readily recognized during infection by the host antibody response. Secondly, we re-addressed a vaccine approach towards the Invariant Surface Glycoprotein ISG75, and showed that despite being highly immunogenic, trypanosomes can avoid anti-ISG75 mediated parasitemia control.

## 1. Introduction

Flagellated salivarian trypanosomes are extracellular single-celled eukaryotic parasites that colonize the mammalian bloodstream, the lymphatics, the brain, and the adipose tissue [1,2,3]. In humans, *Trypanosoma brucei rhodesiense* and *T. b. gambiense* cause sleeping sickness, also known as HAT (human African trypanosomosis) [4]. Animal trypanosomosis (AT) is predominantly caused by *T. congolense*, *T. vivax*, *T. evansi, T. simiae* and, to a lesser extent, *T. b. brucei* [5]. Under normal conditions, none of these AT parasites cause a direct threat to human health as they are unable to grow in the presence of the human serum protein APOL1 [6,7,8]. However, it should be noted that, despite the classification of *T. evansi* as an animal-infective trypanosome, multiple reports have emerged in recent years describing cases of atypical human *T. evansi* trypanosomosis (aHT) in Asia [9,10,11]. In addition, the problem of AT has moved out of the “tsetse belt” and thus spread beyond sub-Saharan Africa. Indeed, in the last century, both *T. vivax* and *T. evansi* have been exported to South America, where the parasites are now migrating towards Central America [12,13,14], and *T. evansi* cases are also reported in Asia and Europe [15,16,17,18,19,20]. This global spread is caused by a combination of intercontinental shipping of infected livestock and mechanical parasite transmission [21,22,23]. Another consideration in the ongoing battle against trypanosomes is that *T. b. rhodesiense* HAT and *T. evansi* aHT are zoonotic infections, meaning they can only be controlled by tackling the vast animal host reservoir. In all trypanosome infections, the anti-parasite response largely relies on the host’s capacity to generate antibodies against trypanosomal antigens. To successfully colonize a broad range of hosts, trypanosomes in turn acquired adaptation mechanisms that allow them to circumvent immune-mediated killing. These have been best studied for *T. brucei* and appear to rely in large on two basic principles, i.e., (i) the use of a variant surface glycoprotein (VSG) coat to evade successful attacks by circulating antibodies, and (ii) the destruction of B cell memory, thereby reducing the need for excessive stringency during the process of antigenic variation [24]. Antigenic variation of the trypanosome VSG coat is considered as a first line of defense against antibody-mediated killing [25,26,27]. The molecular mechanisms that enable VSG variation involve chromosomal recombination, expression site activation/silencing, access to an extensive reservoir of over 1000 genomic VSG genes and pseudogenes, and heterogeneous O-glycosylation of distinct VSGs. Combined, they allow trypanosomes to outrun the mammalian adaptive immune system [28,29]. In addition to antibody evasion, the trypanosome VSG coat also allows active removal of surface binding antibodies and even complement factors through lateral surface movement and endocytosis [30,31,32,33]. The latter is important as IgM-mediated complement cascade activation, CR3-mediated parasite phagocytosis by macrophages, and surface binding of the complement receptor CRIg itself are important in anti-trypanosome clearance [34,35]. However, VSG switching has limitations when it comes to successfully evading immune recognition as (i) structural elements within the surface molecules share conserved T cell epitopes [36], and (ii) invariable surface molecules are required by the parasite for nutrient binding and uptake, as well as structural and motility functions. Hence, to complement the defense against direct surface attacks by the immune system, trypanosomes have invested in mechanisms that allow them to undercut the host’s antibody production capacity by exerting a detrimental effect on the B cell compartment itself [37,38,39,40,41,42]. The latter makes the prospect for successful anti-trypanosome vaccine development challenging, as it requires that the race between the parasite trying to dismantle the immune system, and the immune system trying to eliminate the parasite, should be won by the latter, within days after the trypanosome enters the system. So far, this has not been achieved by any field-applicable approach, despite the publication of multiple promising laboratory results. To address the possibility of anti-trypanosome vaccination for *T. brucei* (and the closely related *T. evansi*), and to illustrate some of the immunological challenges at hand, we tackled two targets, i.e., trypanosomal enolase (ENO), and the invariant surface glycoprotein 75 (ISG75).

Glycolysis is a central metabolic pathway that is highly conserved across all kingdoms of life. Bloodstream form trypanosomes (the predominant parasite form in the mammalian host) use glycolysis as the sole source of ATP production, thereby making this pathway uniquely important for survival. Hence, the glycolysis represents an attractive pathway to target with regards to the development of trypanocidal drugs [43]. Interestingly, in trypanosomes, the first seven reactions of the glycolysis (and the enzymes that catalyze them) are contained within microbody-like organelles called glycosomes [44,45,46,47], which contrasts with most other organisms (in which glycolysis takes place in the cytoplasm). The enzymes catalyzing the last three glycolytic reactions (phosphoglycerate mutase, enolase, and pyruvate kinase) have a cytosolic location and are not part of the glycosomal machinery. This is the case for both *T. b. brucei* and *T. evansi*, two evolutionary related *Trypanozoon* parasites [48] displaying an identical subcellular localization of their glycolytic enzymes [49,50]. ENO is a glycolytic enzyme that catalyzes the reversible conversion of _D_-2-phosphoglycerate to phosphoenolpyruvate. The structural and kinetic features of *T. brucei* ENO have been thoroughly investigated [51,52,53]. A rather unexpected, yet interesting finding, is that trypanosomal ENO is actively secreted by the parasite during infection [54,55,56,57], thereby making it a target for the host immune system. The immunogenicity of the enzyme is highlighted by the fact that unbiased alpaca vaccination with *T. evansi* lysate, followed by phage display panning against the *T. evansi* secretome to obtain camelid single-domain antibodies (sdAbs aka Nanobodies), identified ENO as the most immunogenic target [58]. The presence of ENO in parasite secretome has also been reported for *Leishmania* species, where it has been suggested that an interaction with the host plasminogen receptor plays a role in invasiveness and virulence [59,60,61]. Hence, given its immunogenicity, its accessibility, and its conserved nature, trypanosomal ENO may also be of interest in vaccine research. 

In contrast to ENO, ISG75 is a membrane-anchored, trypanosomal surface antigen identified three decades ago. Meanwhile, ISG75 has been identified as the surface binding target of the anti-trypanosomal drug suramin [62,63,64], although its exact biological function remains elusive. Despite the reported failure of ISG75 to induce a host antibody response in chronically infected animals [65], it was nonetheless assessed as a vaccine candidate due to its surface-exposed nature [66,67]. These initial studies did not show any protective efficacy of anti-ISG75 vaccination, and it was assumed that the antigen was rendered inaccessible to host antibodies by the trypanosome’s VSG coat. However, this hypothesis was called into question as more recent reports demonstrated that ISG75 can in fact be targeted by the host antibody response. In combination with *T. b. gambiense* VSG LiTat 1.5, ISG75 is one of the most promising specificity-increasing markers for HAT antibody-based diagnostics [68,69,70] and has also been shown to be a suitable target for *T. evansi* diagnosis in bovines [71,72]. Other work has shown that antibody-ISG75 surface interactions can be used to study membrane recycling process in trypanosomes, as well as for the localization of ISGs on the trypanosome surface [73,74]. Importantly, the protective effect of an experimental trypanosome flagellar pocket vaccine approach coincided with the presence of anti-ISG75 antibody induction [75].

In this paper, recombinant *T. brucei* ENO (*Tb*ENO) and *T. b. gambiense* ISG75 (*Tbg*ISG75) were thoroughly examined as vaccine candidates in *T. b. brucei* and *T. evansi* infection models. The rationale behind including trypanosomal ENO in the study stemmed from the idea that trypanosomes would only have only evolved to actively secrete one of their glycolytic enzymes if there would be a biological benefit. Hence, we wondered whether host antibodies responses against secreted trypanosomal ENO would have a detrimental effect on parasite fitness. ISG75 was selected in order to re-assess the potential of ISG75 as an immunogen, considering the more recent data with regards to ISG75-host antibody interactions. Our results demonstrate that both *Tb*ENO and *Tbg*ISG75 are exposed to the host humoral immune response and are highly immunogenic, as specific antibodies can be easily induced in experimental mouse models. However, these antibody responses are non-protective and have no impact on parasitemia when immunized mice are challenged with parasites.

## 2. Materials and Methods

### 2.1. Mice and Parasites Infections

Female 7- to 9-week-old C57Bl/6 were purchased from Koatech (Gyeonggy-do, South Korea). Mice were infected with the pleomorphic *T. b. brucei* AnTat 1.1E (EATRO 1125 stock) through intraperitoneal (i.p.) injection of 5000 parasites/mouse [68]. Cryo-stabilates containing 50/50 blood/Alsever (Sigma, St. Louis, MO, USA) with 10% glycerol were thawed and parasite counts were determined by hemocytometer. Samples were diluted in serum-free DMEM medium (Capricorn Scientific, Ebsdorfergrund, Hessen, Germany) to 10^4^ parasites mL^−1^ and 100 μL of the dilution was used to infect the mice. For *T. evansi* infections the ITMAS 150799 ‘Colombia’ stabilate was used. This stabilate produces a reproducible infection pattern similar to that of *T. b. brucei* AnTat 1.1 when injected at a concentration of 200 parasites (i.p.) per mouse. Parasite inoculum sample dilution was done as described above. Every 2 to 3 days, blood parasite numbers were determined using a hemocytometer and light microscope and a 2.5 μL tail vein blood sample diluted 1/200 in DPBS (Invivogen, San Diego, CA, USA). Parasitemia was recorded for a total period of 38 days (*T. brucei*) or 50 days (*T. evansi*), to maximum survival time of both infections, respectively. All experimental mouse procedures were approved by the GUGC Institutional Animal Care and Use Committee (IACUC) (file # 2019-026).

### 2.2. Recombinant Production and Purification of TbENO

The recombinant production of *Tb*ENO in *E. coli* BL21(DE3) and its subsequent purification through a two-step protocol encompassing immobilized metal ion affinity and size exclusion chromatography (IMAC and SEC, respectively) were performed as previously described [58].

### 2.3. Recombinant Production and Purification of TbgISG75

The gene fragment encoding the extracellular domain of *Tbg*ISG75 (*T. b. gambiense* LiTat 1.3, accession number DQ200220.1) comprising residues 29–462 of the full-length protein was cloned into the mammalian expression vector pHLsec [76]. It encodes an N-terminal secretion sequence and a thrombin-cleavable, C-terminal hexahistine-tag. Transfection of Expi293F cells (ThermoFisher, Waltham, MA, USA) was carried out according to manufacturer instructions. Briefly, 100 mL of exponentially growing cells were transfected with 100 µg of plasmid. Kifunensine was added to the cell suspension at a concentration of 5 µM immediately after the transfection to produce a homogenous glycosylation pattern. Cells were harvested 4 days after transfection by centrifugation (10 min, 2000× *g*, 20 °C). The supernatant was filtered using a 0.45 µm bottle top filter unit and dialyzed twice for at least 4 h against 20 volumes of IMAC equilibration buffer (20 mM Tris, 500 mM NaCl, 10 mM Imidazole, pH 8.0). The dialyzed cell supernatant was applied to Nickel-NTA agarose beads (Qiagen, Hilden, Germany) using a gravity-flow column (BioRad, Hercules, CA, USA), washed with 20 column volumes of IMAC wash buffer (20 mM Tris, 500 mM NaCl, 30 mM Imidazole, pH 8.0) and the bound protein was eluted with 5 column volumes of IMAC elution buffer (20 mM Tris, 500 mM NaCl, 400 mM Imidazole, pH 8.0). Cleavage of the His-tag was carried out by addition of thrombin (GE) to the eluate which was immediately dialyzed against IMAC equilibration buffer at 20 °C overnight. Cleaved protein was separated from uncleaved protein by applying it again to nickel beads. The flow-through was subsequently concentrated to 500 µL using an Amicon centrifugal ultrafiltration device (Merck Millipore, Burlington, MS, USA) with a 10 kDa molecular weight cut-off (3300× *g*, 4 °C) and injected onto a Superdex S200 10/300 increase size exclusion column (GE healthcare) previously equilibrated with SEC running buffer (20 mM HEPES, 150 mM NaCl, pH 7.5). The fractions containing *Tbg*ISG75 were pooled, flash-frozen in liquid N_2_ and stored at −80 °C. Each purification step was monitored by SDS-PAGE for absence of contaminations, as well as degradation and >95% purity of the target protein in the final step. 

### 2.4. Circular Dichroism (CD) Spectroscopy

For *Tb*ENO, far-UV CD data were recorded on a J-175 spectropolarimeter (Jasco, Easton, MD, USA). Continuous scans were taken using a 1 mm cuvette, a scan rate of 50 nm min^−1^, a band width of 1.0 nm, and a resolution of 0.5 nm. Five accumulations were taken at 20 °C in 50 mM sodium phosphate, 5 mM MgCl_2_, pH 7.0 at a protein concentration of 0.2 mg mL^−1^. Thermal unfolding experiments were performed by gradually increasing the temperature from 10 to 90 °C at a constant rate of 1 °C min^−1^. To follow the change in α-helicity, the mean residue ellipticity measured at 222 nm was plotted as a function of the temperature.

For *Tbg*ISG75, far-UV CD experiments were carried out on a Jasco J-1500 spectropolarimeter with a 0.2 mm path cell. *Tbg*ISG75 was measured at a concentration of 0.4 mg mL^−1^ in 10 mM Hepes, 150 mM NaF, pH 7.5. Spectra were recorded between 195 and 280 nm wavelength at an acquisition speed of 10 nm min^−1^ and corrected for buffer absorption. For calculation of melting curves at 222 nm, spectra were recorded every 5 degrees between 5 °C and 95 °C with a slope of 0.16 °C min^−1^.

In both cases, the raw CD data (ellipticity θ in mdeg) were normalized for the protein concentration and the number of residues according to the equation below, yielding the mean residue ellipticity ([θ] in deg cm^2^ mol^−1^), where *MM*, *n*, *C*, and *l* denote the molecular mass (Da), the number of amino acids, the concentration (mg mL^−1^), and the cuvette path length (cm), respectively.
(1)$$\left[\theta \right]=\frac{\theta \cdot MM}{n\cdot C\cdot l}$$

Secondary structure content was calculated using the DichroWeb server [77].

### 2.5. Small-Angle X-ray Scattering (SAXS) Data Collection and Processing

SAXS data were collected at ESRF, Grenoble (France), SAXS beamline BM29, using a Pilatus 2M detector (DECTRIS) and a wavelength of 0.99 Å at 20 °C. For SEC-SAXS, 50 µL ISG75 at 9.8 mg mL^−1^ were injected onto a Superdex 200 3.2/300 equilibrated in 20 mM Hepes, 150 mM NaCl, 3% (*v*/*v*) glycerol, pH 7.5 at a flow rate of 75 µL min^−1^. Scattering data were acquired as components eluted from the column and passed through the SAXS measuring cell. The ATSAS [78] software package was used to normalize the data to the intensity of the incident beam, average frames and subtract scattering contribution from the buffer. In detail, 10 frames corresponding to the void volume of the column were averaged and subtracted from ten averaged frames of the main elution peak. The radius of gyration, *R_g_*; the maximum particle dimension, *D_max_*; and the distance distribution function, *p*(*r*), were evaluated using the program PRIMUS as part of the ATSAS package [78]. The excluded volume of the hydrated particle (the Porod volume, *V_p_*) was computed using the Porod invariant [79]. The particle mass was calculated using the Bayesian interference method [80]. Particle shapes at low resolution were reconstructed ab initio by the bead-modeling program DAMMIF [81] using default parameters and evaluated using their normalized spatial discrepancy parameter. UCSF Chimera [82,83] was used for envelope reconstruction.

### 2.6. Isothermal Titration Calorimetry (ITC)

The interactions between *Tb*ENO and anti-*Tb*ENO camelid single-domain antibodies (sdAb11 and sdAb77) [58] were investigated by ITC on a MicroCal PEAQ-ITC instrument (Malvern). In all experiments, the sdAbs were titrated into the sample cell containing *Tb*ENO. The following concentrations were used for the different data sets: sdAbs (50 μM)-*Tb*ENO (5 μM). All proteins were extensively dialyzed against the same buffer (50 mM sodium phosphate, 5 mM MgCl_2_, pH 7.0) to exactly match buffer composition. Before being examined in the calorimeter, all samples were degassed for 10 min at a temperature close to the titration temperature (25 °C) to prevent long equilibration delays. The reference power was set to 5 μcal s^−1^ and a stirring speed of 750 rpm was used. An equilibrium delay of 180 s before the start of each measurement was employed, while a spacing of 150 s between each injection was used. During data collection, sixteen injections were used with variable injection volumes ranging from 1.5 to 3.0 μL. The first injection was always 0.4 μL and its associated heat was never considered during data analysis. To determine the injection heats, control titrations were performed consisting of injections of sdAbs into the buffer-filled cell (thus in the absence of *Tb*ENO). Baseline adjustment, control subtraction and data analysis were performed using the MicroCal PEAQ-ITC analysis software. The data were analyzed with the “one set of sites” binding model resulting in fitted values for the stoichiometry of the interaction (N), the association constant (K_a_), and the change in enthalpy (ΔH_a_) and entropy (ΔS_a_) associated with the binding events. All experiments were performed in triplicate.

### 2.7. Vaccine Procedure and In Vivo T. b. brucei Challenge

Mice were vaccinated according to the previously published protocol [75]. In short, vaccination was performed using 10 µg recombinant protein per mouse, emulsified in 100 µL DPBS (Invivogen, San Diego, CA, USA) and 100 µL Complete Freunds’ Adjuvants (CFA, Sigma) administered subcutaneously (s.c.) in the scruff of the neck. After 21 and 42 days, mice received a booster injection (s.c.) with the same amount of recombinant protein, but substitution CFA by IFA (incomplete Freunds’ Adjuvant, Sigma). Twelve weeks later, mice were infected i.p. with either 5000 *T. b. brucei* parasites/mouse or 200 *T. evansi* parasites/mouse. Mock vaccinated mice received non-supplemented adjuvants injections following the same injection scheme.

### 2.8. Quantification of Anti-TbENO and Anti-TbgISG75 Antibody Titers by ELISA

*Tb*ENO or *Tbg*ISG75 was coated on a 96-well half-area clear flat bottom polystyrene high-bind microplate (Corning Inc., Corning, NY, USA) at 4 °C overnight, using 0.1 µg/50 µL/well in 0.05 M bicarbonate 9.6 pH coating buffer (3.7 g sodium bicarbonate (NaHCO_3_)/0.64 g sodium carbonate (Na_2_CO_3_)/1 L H_2_O). Heparinized plasma was collected from infected mice and stored in aliquots at −20 °C. At several timepoints throughout the experiment, plasma IgM, IgG1, IgG2b, IgG2b and IgG3 titers were determined using a plasma diluted series of 1/3, starting at 1/200, with DPBS (Invivogen, San Diego, CA, USA) as diluent and horseradish peroxidase-labeled specific secondary antibodies (Southern Biotech, Birmingham, AL, USA, kit Cat. No. 5300-05). TMB (Sigma, St. Louis, MO, USA Cat. No. T0440) substrate conversion was measured at 450 nm (with 570 nm as background wavelength) using a Multiskan *PC* ELISA reader (Thermo Scientific, Waltham, MA, USA). For all ELISA assays, graphic representations of the results were generated using a non-linear regression with asymmetrical five-parameter curve fitting (GraphPad, San Diego, CA, USA). 

### 2.9. Cell Preparation and Flow Cytometry Analysis

Spleen cells were isolated at 14 days post infection (dpi). Homogenized single-cell suspensions were prepared in 6 mL of DMEM (Capricorn Scientific, Ebsdorfergrund, Hessen, Germany) supplemented with 10% FBS (Atlas Biologicals, Fort Collins, CO, USA) and 1% penicillin/streptomycin using gentleMACS™ Dissociator (Miltenyi Biotec, Bergisch Gladbach, Germany). Cells were passed through a 70-µm cell strainer (SPL Life Sciences, Gyeonggi-do, South Korea), and centrifuged at 314× *g* for 7 min at 4 °C, followed by re-suspension and incubation in RBC lysis buffer (BioLegend, San Diego, CA, USA) at 4 °C for 5 min. After washing (314× *g* 7 min at 4 °C), cells were kept on ice in FACSFlow sheath fluid (BD Biosciences, San Jose, CA, USA) containing 0.05% FBS (Atlas Biologicals, USA) and Fc block (CD16/CD32 Fcγ III/II, BioLegend, San Diego, CA, USA) (1/1000 dilution) for 30 min in the dark at 4 °C. Next, 10^5^ cells per sample were incubated for 30 min in the dark at 4 °C, with antibody cocktails specific for different splenocytes populations, followed by flow cytometry analysis using a BD Accuri™ C6 Plus flow cytometer (BD Biosciences, San Jose, CA, USA). The gating strategies used have previously been published [84], and the percentage of each population was determined by dividing the number of events within a specific gate, by the total number of events within live gate.

### 2.10. Flow Cytometry Detection Reagents 

The following antibodies (BioLegend, San Diego, CA, USA) were added to 100 µL aliquots of 10^5^ Fc-blocked splenocytes prepared as described above to make a final 1/600 dilution: anti-CD1d-PE(clone 1B1), anti-B220-FITC (clone RA3-6B2), anti-CD93-APC (clone AA4.1), anti-CD138-PE-Cy7 (clone 281-2), anti-GL7-PE (clone GL7), anti-Ly6G-Alexa488 (clone 1A8), anti-Ly6C-PE (clone HK 1.4), anti-CD4-FITC (clone GK 1.5), anti-CD8-PE (clone 53-6.7), anti-NK1.1-APC(clone PK 136), anti-TCRβ-PE-Cy7 (clone H57-597).

### 2.11. Statistical Analysis

GraphPad Prism v.8.3 (GraphPad Software Inc., San Diego, CA, USA) was used for data presentation and statistical result analysis. Data were compared with naïve samples using Student’s *t*-test. Means are given as ±standard deviation (SD).

## 3. Results

### 3.1. Recombinant TbENO and TbgISG75 Are Correctly Folded and Biologically Relevant

The *Tb*ENO and *Tbg*ISG75 immunogens employed in this study were obtained from recombinant sources. To ensure that the target proteins are properly folded and thus biologically relevant, they are subjected to a stringent quality control relying on biochemical and biophysical methods.

*Tb*ENO is recombinantly produced in *E. coli* BL21(DE3) and subsequently purified from the cytoplasmic fraction via a two-step purification protocol encompassing IMAC and SEC (Figure 1A). We have previously demonstrated that this approach yields highly pure, enzymatically active, dimeric, and properly folded *Tb*ENO batches [58]. To underline the reproducibility of our methods, we again performed CD spectroscopy. As can be seen from Figure 1B, the CD spectrum of *Tb*ENO contains the typical features of a protein composed of α-helices and β-sheets, which is in line with the expectations based on the *Tb*ENO crystal structure [52]. Thermal denaturation followed at 222 nm reveals an apparent melting temperature (T_m,app_) at ~54 °C, which is in accordance with our previous findings [58] and supports the presence of a properly folded quaternary structure pre-transition. In addition, we investigated the interactions between *Tb*ENO and two anti-*Tb*ENO sdAbs (sdAb11 and sdAb77) by ITC. Both sdAbs were obtained previously and recognize the native antigen with high affinity and specificity [58]. The ITC data reveal that both sdAbs bind recombinant *Tb*ENO with high affinity and with the previously reported stoichiometry of one sdAb for one *Tb*ENO monomer (Figure 1C), thereby strongly advocating that recombinant *Tb*ENO is equivalent to its native counterpart secreted by the parasite. 

*Tbg*ISG75 is recombinantly produced in transiently transfected mammalian cells (HEK293F) from which it is secreted into the culture medium. From here, *Tbg*ISG75 is purified to homogeneity through a combination of IMAC and SEC (Figure 1A). As for *Tb*ENO, we employed CD spectroscopy to assess the presence of secondary structure elements (Figure 1D). *Tbg*ISG75 shows features typically found in surface proteins of African trypanosomes such as an all α-helical fold and a high content of low-complexity regions (LCRs; turns and random coils) [85]. In comparison, the structure of a typical VSG (PDB-ID: 2VSG [86]) consists of 52.0% alpha helices, 5.3% beta sheets, 23.4% turns and 19.3% random coils. The structure of the *Trypanosoma brucei* transferrin receptor (PDB-ID: 6SOZ [87]) representative for other surface proteins has 54.8% percent alpha-helical content, 4–6% beta-sheets, 22.3% turns and 18.3% random coils. LCRs in trypanosome surface proteins are located mainly in the membrane distal head domain where they can act as immune decoys or ligand-binding regions as well as the membrane-proximal, C-terminal domains that connect to the GPI-anchor (or in case of ISG75 to the transmembrane domain) and create a flexible cushion between the N-terminal domains and the cell membrane [88]. In addition, *Tbg*ISG75 displays a relatively high thermostability with a T_m,app_ of ~41 °C and a clear folding transition, suggestive of tertiary structure (Figure 1D). Small-angle X-ray scattering (SAXS) experiments further confirm that recombinant *Tbg*ISG75 preparations are indeed monodisperse, containing well-folded, moderately elongated particles with dimensions typical for the canonical 3-helix bundle fold of trypanosomal surface proteins [8,89] (Figure 1E).

Altogether, the data demonstrate that the obtained recombinant protein batches are of high purity and quality. The ENOs and ISG75s from *T. brucei* and *T. evansi* display a sequence identity of 100.0% and 95–98%, respectively, thereby justifying their use as immunogens in both *T. brucei* and *T. evansi* experimental mouse vaccination and infection models.

### 3.2. TbENO Is a Highly Immunogenic Antigen When Used in a CFA/IFA Vaccination Protocol

Given the presence of trypanosomal ENO in the parasite’s secretome and its potential role in host–parasite interactions, we investigated its immunogenicity. Recombinant *Tb*ENO was employed to vaccinate C57BL/6 mice three times with 3-week intervals between each injection, with immunogens emulsified in a 1/1 protein (PBS)/CFA (first vaccination) or 1/1 protein (PBS)/IFA preparation (2nd and 3rd boost) (Figure 2A). Three weeks after the third boost, specific antibodies titers were measured focusing on class-switched IgG responses. The obtained data indicate that *Tb*ENO is highly immunogenic with the following endpoint titers (Figure 2B): (i) circulating IgG1 exceeding 1/10^6^, (ii) IgG2b and IgG2c levels close to 1/500,000, and (iii) IgG3 reaching just over 1/10^4^. Mice were kept unchallenged for a total of 12 weeks to ensure that a subsequent parasite challenge was not affected by the immediate presence of acute vaccine-induced immune responses. Prior to challenge with parasites, the endpoint titers of all IgG anti-*Tb*ENO subclasses were measured once again and showed a 3-fold dilution drop for IgG1, IgG2b and IgG2c in endpoint titers, while a 9-fold drop was recorded for IgG3. These results are indicative of the circulation of vaccine-induced long-lived plasma cells. Twelve weeks after the third boost, mice were challenged with *T. b. brucei* or *T. evansi* parasites and at 14 dpi (days post infection), specific anti-*Tb*ENO endpoint titers were determined once more. The data in Figure 2B show that *T. brucei* infection causes a marginal increase in IgG2c and IgG3 titers (i.e., a one-dilution shift in the 3-fold dilution series), while IgG1 and IgG2b remain the same as those observed prior to infection. For the *T. evansi* infection, no measurable differences in endpoint titers were observed before and after infection.

### 3.3. TbgISG75 Is a Highly Immunogenic Antigen When Used in a CFA/IFA Vaccination Protocol

Because of the ongoing debate with regards to whether ISG75 can efficiently elicit host antibodies, we re-assessed the immunogenicity of *Tbg*ISG75 by employing the same experimental setup outlined above for *Tb*ENO (Figure 2A). Here, our data show that *Tbg*ISG75 is immunogenic, albeit slightly less when compared to *Tb*ENO, with the following endpoint titers (Figure 3): (i) circulating IgG1 levels of 1/500,000, (ii) IgG2b and IgG2c levels just over 1/10^5^, and (iii) significantly lower IgG3 levels of 1/2000. Furthermore, here, mice were kept unchallenged for a total of 12 weeks, during which IgG endpoint tires dropped 3- to 9-fold, with the larger drop for IgG1 and IgG2c. Twelve weeks after the third boost, mice were challenged with either *T. b. brucei* or *T. evansi* parasites and at 14 dpi, specific anti-*Tbg*ISG75 endpoint titers were determined. The data in Figure 3 show that *T. brucei* challenge had no effect on the endpoint titers of IgG1 and IgG2b, induced a 3-fold induction of IgG2c and also caused a 9-fold increase in IgG3 anti-*Tbg*ISG75 titers. For the *T. evansi* infection, a similar trend was noted albeit the increase in IgG3 was less pronounced (3-fold).

### 3.4. TbENO/TbgISG75 Vaccination Does Not Alter T. b. Brucei or T. evansi Infection Progression

Following vaccination and a 12-week waiting period, mice were challenged with either 5000 *T. b. brucei* AnTat 1.1 parasites, or 200 *T. evansi* ‘Colombia’ parasite by i.p. injection. For both groups, 10 mice were used, and parasitemia was monitored at 2-day intervals. Figure 4A shows that neither the *Tb*ENO nor the *Tbg*ISG75 vaccination had any measurable effect on parasitemia development of the clonal *T. b. brucei* AnTat 1.1 parasite. The first peak of infection occurred at the same time in all mice. Mice were euthanized as they developed severe end-stage anemia and infection-associated morbidity, which happened in a similar timeframe in the three experimental groups. Similarly, the *T. evansi* ‘Colombia’ stabilate developed a parasitemia profile that was similar in all experimental groups (Figure 4B). As this stabilate is close to its field origin, the parasitemia following the first peak showed grater variations than those observed in the *T. b. brucei* infection, but overall, there was no difference in survival between the mice in the three experimental groups.

### 3.5. TbENO/TbgISG75 Vaccination Does Not Alter T. b. brucei Infection-Associated Immunopathology

Inflammatory modulation of the spleen compartment is a hallmark of experimental trypanosomosis. Hence, the splenic cellular composition of naïve mice was compared to the compositions found in both *Tb*ENO and *Tbg*ISG75 vaccinated mice, at 12 weeks after the third boost, as well as in vaccinated infected mice. Finally, these observations were compared to the splenic cellular composition of non-vaccinated infected mice. These experiments focused on the *T. b. brucei* AnTat 1.1 infection and confirmed the lack of any positive vaccine-related outcome on infection. The experiments were not repeated for the *T. evansi* ‘Colombia’ stock. In order to identify mature B cell subpopulations, spleen cells were first gated based on their FSC/SSC pattern and subsequently selected based on B220^+^ expression, while excluding CD138^+^ and CD93^+^ cells representing plasma cells and immature B cells, respectively (Figure 5). Plasma cells were directly gated from the FSC/SCC plot, using B220^+^/CD138^+^ as phenotype. Monocytes were also characterized directly from the FSC/SCC plot, using a CD11b^+^/Ly6C^+^ plot to visualize double-positive cells, while granulocytes were identified using a CD11b^+^ pre-gating, followed by a Ly6G^+^/Ly6C^+^ plot, in which the Ly6G^+^ population is clearly identifiable. For both marginal zone and follicular B cells (MZB/FoB), it is clear that both naïve and vaccinated mice have similar population sizes, but that infection has a dramatic effect after two weeks, leading to the total collapse of the MZB compartment, and a significant reduction of 50–75% in FoB cells population size. This reduction is seen in both vaccinated and non-vaccinated infected mice, showing that there is no beneficial effect noticeable that could be attributed to either the *Tb*ENO or *Tbg*ISG75 immunization strategy. The plasma cell population was low in both naïve and vaccine mice (0.19–0.34% of the total spleen) but increased dramatically upon infection (7.89–10.42%). However, also here the effect is mainly attributed to infection itself. Interestingly, the data in Figure 5 suggest that the increase in plasma cells is slightly higher in the *Tb*ENO-vaccinated mice, coinciding with the slight infection-induced increase in anti-*Tb*ENO IgG titers in vaccinated mice presented in Figure 2. The slight increase in plasma cells (PCs) in the infected *Tbg*ISG75 vaccinated mice coincided only with a slight increase in IgG2c infection-induced antibody titers. Interestingly, in both vaccine groups, the percentage of spleen monocytes increased at least 2-fold as compared to the non-vaccine naïve mice. This increase is, however, dwarfed by the infection-induced 3- to 6-fold expansion of this population in infected mice. Here also expansion seems to be mainly driven by infection itself, as there was no difference observed between non-vaccinated infected mice and *Tbg*ISG75-vaccinated infected mice. In *Tb*ENO-vaccinated mice there appeared to be a more pronounced accumulation of spleen monocytes, but this obviously did not have a beneficial effect for parasitemia control. Finally, splenic granulocyte accumulation has been identified in the past as a hallmark of trypanosomosis-associated spleen inflammation, a finding that is confirmed here. As for the monocytes, it appears that both the *Tb*ENO and *Tbg*ISG75 immunizations resulted in a prolonged 2-fold increase in the presence of spleen granulocytes. However, at 14 dpi, the further increase in the granulocyte population size (most notably in the *Tb*ENO-vaccinated group) showed that none of the mice showed any level of protection against infection-induced immunopathology and inflammation.

## 4. Discussion

Vaccination is considered to be the key to success when it comes to controlling the spread of infections, as well as limiting the detrimental health effect of diseases [90,91]. While the last century has shown that vaccines can eradicate disease such as smallpox and polio, and the recent Covid-19 pandemic has shown how vaccines can reduce disease-associated death and morbidity, successful examples are mostly related to fighting viral and bacterial infections [92]. In contrast, when it comes to vaccine efficacy against parasites, progress has been slow or unsuccessful. While a popular notion could exist that the latter is due to lack of funding and/or scientific interest, the reality is different. Indeed, when dealing with parasites, one must consider that these organisms have adapted to live within the defense environment of their host. For parasites to be able to do so, they must be perfectly adapted to overcome all the defense systems of the innate and adaptive immunity, as their ultimate goal is to cause long persistent infections without necessarily causing harm to their host. In the end, there appears to be no evolutionary advantage for a parasite to kill its host through excessive pathology. 

Within the group of mammalian parasites, salivarian trypanosomes must be considered as rather unique, as they have adopted all necessary means to survive in plain sight of the host immune system [24]. They are extracellular single-cell organisms that should in principle be easily targeted by the phagocytic as well as the antibody-mediated immune system, yet they thrive in the blood and lymphatics of their host, and cause infections that can perpetuate for years. Trypanotolerant animals, such as N’Dama cattle and West African dwarf goats and sheep, can indeed serve as transmission hosts for both HAT and AT, without suffering from severe infection-associated pathology themselves. Interestingly, trypanosomes have always been at the forefront of medical science, ever since Paul Ehrlich conceptualized the idea chemotherapy, by using the cellular dyes trypan red and trypan blue as trypanotoxins [93]. Much later, trypanosome immunology was part of the discovery of cytokine function, with the study of *T. b. brucei*-induced inflammation and AT-associated wasting disease being at the cradle of the identification of cachectin/TNF [94,95]. Finally, in the field of molecular biology and antigenic variation, trypanosomes have offered a unique research model since the discovery of the VSG coat in 1975 [96,97,98]. Unfortunately, it was also the discovery of VSG switching which led Cornelissen et al. to conclude in 1985 that the prospect for the development of a successful field applicable anti-trypanosome vaccines was “not good” [99], a statement that still stands today. With the additional discovery that trypanosomes adopt a very active strategy of B cell and memory destruction, the situation looks even more complicated [41]. Obviously, the latter strategy prevents attack by cross-reacting antibodies, or the T-cell mediated recognition of conserved VSG epitopes, when different but similar VSG are being used in successive parasitemia waves. The latter has been shown to occur, even across trypanosome species [100]. Unfortunately, the infection-induced B cell destruction also causes collateral damage, as it results in the abrogation of immunization efficacy of non-related vaccines in trypanosome-infected animals, and possibly humans [101]. 

In recent years, at least 15 publications have shown ‘promising’ vaccine results in various trypanosome models, albeit most of them under laboratory conditions [101]. Unfortunately, virtually none of them addressed the immunological aspect of memory recall activation [102]. Strategies included intracellular targets inaccessible to antibody attack (such as tubulin) and implemented most often extremely short waiting periods between boost and challenge, making it hard to grasp which immunological principle would have been triggered to provide protection. A recent study on the use of the *T. evansi* paraflagellar rod proteins PFR1 and PFR2, in which the infection challenge was performed only 7 days after the last vaccine boost, showed that the presence of high anti-target IgG titers at the moment of parasite challenge could delay the onset of parasitemia, but in the end failed to provide any sterile protection [103]. Most recently, it was shown that an anti-*T. vivax* vaccine using the ectodomain of the invariant cell-surface flagellum antigen IFX conferred protection against a subsequent *T. vivax* challenge in 10/15 mice, and that protection was observed even when parasite challenges were performed repeatedly over a 100-day period following the last vaccine boost. Unfortunately, also here, preliminary results showed that the protective effect observed in mice could not be translated to a goat model, possibly because mice were challenged with a rather low dose of parasites [104]. 

The trypanosomal flagellar pocket (a membrane invagination concentrating ISGs, including ISG75, and other receptors), appeared at one stage to be a good lead to build an anti-trypanosome vaccine approach. Anti-FP vaccination provided partial protection against both heterologous experimental infections in mice, as well as natural challenges in cattle [75,105]. Importantly, the controlled laboratory setup showed that the infectious dose was crucial, and that protection was only observed with a low-dose challenge, i.e., 500 parasites. Considering that a tsetse fly bite is believed to contain up to 40.000 metacyclic trypomastigotes [106], this could explain why translation of laboratory anti-trypanosome vaccination results to field settings have not been successful so far. Hence, in the study performed here, mice were challenged with 5000 parasites (a standard dose in mouse *T. b. brucei* research [68]) to potentially avoid inserting a favorable bias towards the outcome in the experimental design. In addition, the time between the last vaccine boost and parasite challenged was chosen to be extended up to 12 weeks. The rationale here was to test vaccine memory, rather than the immediate effect of high titers of vaccine-induced antibodies. During this ‘waiting’ period, mice did not receive additional exposure boosts with parasite material, as was done in the *T. vivax* vaccination approach referred to above [104], since this would have biased the immune system to maintaining increased anti-target antibody titers. Hence, using our approach, we have shown that the recombinantly obtained *Tb*ENO and *Tbg*ISG75 are highly immunogenic molecules, capable of inducing extremely high IgG antibody titers in a CFA/IFA vaccination protocol. Despite a strong humoral response, these immunogens were unable to confer protection against either *T. b. brucei* or *T. evansi* in a proper memory recall/challenge experiment. The stringent quality control imposed on our recombinant protein batches ensure that they are of high purity and quality and eliminate the possibility that vaccine failure was the result of poorly folded, low-quality, low-purity immunogens. Instead, the obtained results most likely once more point to the fact that a molecular arms race starts as soon as trypanosomes are exposed to the immune system. On one hand, the parasite needs to undermine the specific antibody production capacity by the host, while on the other, the host immune system aims to kill the parasite before successful VSG switching starts to take place. In the case of *T. b. brucei*, it has been described that the detrimental effect on the B cell compartment is initiated 6 days post infection [41], and the results outlined here confirm that such destructing takes place, also during *T. evansi* infections. Hence, these data show that trypanosomes win the arms race, as the host simply seems incapable of turning any B cell memory into a fully activated plasma cell antibody production compartment within the first 3–4 days of infection. The mammalian immune system simply needs more time, and the trypanosome apparently has adapted to the timing constraints of the mammalian immune system. Secondly, it is well possible that the mechanisms of antibody-VSG ‘sailing’ that have been described as an immune-evasion mechanism to clear host antibodies from the surface, also reduce surface anti-ISG75 levels below a critical threshold level [30,31,107]. As for the lack of vaccine effect in the ENO setup, it is possible that the enzyme plays a role in natural transmission cycle of infection (which was not addressed here) or that the released form of the enzyme plays a biological role in the host, in a location where it is inaccessible to vaccine-induced antibodies (for example, inside host cells).

With the recent success in the control of HAT [108], one might wonder whether there is still a need for continuing anti-trypanosome vaccine research. However, the final battle against HAT cannot be won without remembering (i) that the vast wildlife animal reservoir cannot be controlled by active case detection and treatment, and (ii) that farm animals function as intermediate hosts for HAT transmission. Hence, anti-trypanosome vaccination of livestock is the only viable approach to the sustainable control of trypanosomosis, preventing re-emergence of HAT, and eliminating the economic hardship created by AT for smallholder farming communities. Therefore, we would like to conclude that it is important to continue the research towards functional anti-trypanosome vaccines that will make a difference in the field. It could even be that in highly endemic areas, long-lasting immunological memory responses are less important, as regular natural exposers would possibly keep vaccine-induced IgG responses high enough to exert their critical function. With respect to the choice of antigens, maybe it is time to take a new approach. While deep-learning B cell epitope prediction certainly has its merits when it comes to diagnostic developments [109], the idea that this would lead the way into new vaccine developments [110] is however doubtful. This conclusion is based on the fact that (at least in the case of trypanosomosis) the problem at hand seems to be unrelated to the lack of immunogenicity of target molecules. Hence, the ‘alternative’ answer for future anti-trypanosome vaccine development should maybe be sought in looking for non-immunogenic proteins, such as the *T. vivax* IFX candidate [104]. This target, representing the ectodomain of a newly identified invariant cell-surface trypanosome protein did induce sterile protection against the experimental challenge with trypanosomes. Indeed, if there is one chance to catch trypanosomes ‘off guard’, it should most likely be done by providing a potential future host with the capacity to load the plasma with antibodies against conserved trypanosome targets prior to a first challenge. Such targets would obviously only be conserved when trypanosomes did not experience the evolutionary need to build in variation. Mounting an artificial IgG response prior to exposure hence would give the host a head start in the above-mentioned molecular arms race, beating the trypanosome in its race to destroy the host B cell immune response. Unfortunately, convincing granting agencies of the logic behind deliberately building a vaccine research program starting with non-immunogenic targets might prove to be difficult. The future will tell.

## Figures and Tables

**Figure 1 pathogens-10-01050-f001:**
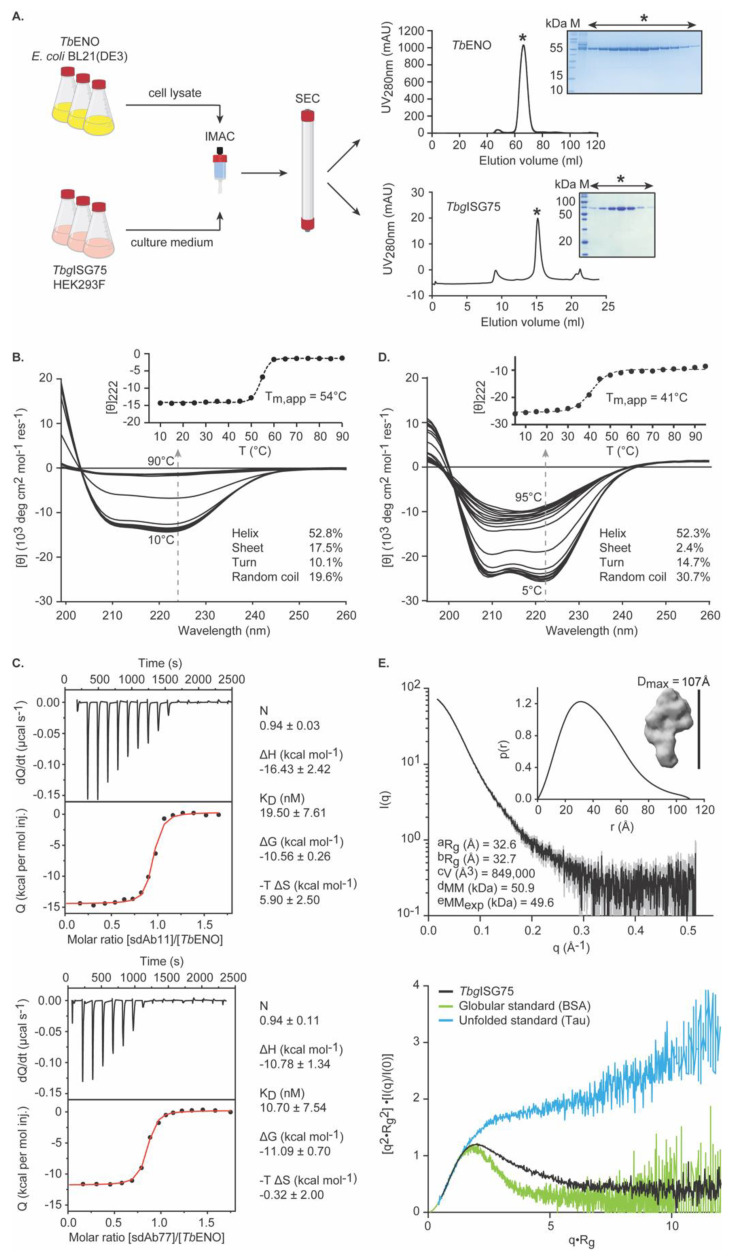
Recombinant *Tb*ENO and *Tbg*ISG75 are properly folded immunogens. (**A**) Schematic representation of the flowchart for recombinant protein production and purification. Typical gel filtration profiles are shown for both proteins (performed on S200 16/60 and S200 10/30 columns for *Tb*ENO and *Tbg*ISG75, respectively). The inset represents an SDS-PAGE analysis of the fractions collected for the elution peak (indicated by ‘*’). *Lane M*, molecular weight marker. (**B**) CD spectra of *Tb*ENO collected at different temperatures. The gray arrow represents the effect of the increasing temperature on the mean residue ellipticity measured at 222 nm, plotted in the inset (filled circles and dashed line represent the experimental data points and fit, respectively). Below 50 °C, spectra show features that are typical for proteins containing α-helices and β-sheets. The secondary structure content calculated based on the experimental data is also shown for convenience. (**C**) ITC measurements at 25 °C for the binding of sdAb11 and sdAb77 to *Tb*ENO. The top panels represent the thermograms in which the black lines depict the raw data. The bottom panels show the isotherms. The black dots display the experimental data points, and the red traces show the fit. The experimentally determined thermodynamic parameters related to the binding events are shown. (**D**) CD spectra of *Tbg*ISG75 collected at different temperatures. The gray arrow represents the effect of the increasing temperature on the mean residue ellipticity measured at 222 nm, plotted in the inset (filled circles and dashed line represent the experimental data points and fit, respectively). Below 40 °C, spectra show minima at 208 and 222 nm that are characteristic for α-helical proteins. The secondary structure content calculated based on the experimental data is also shown for convenience. (**E**) *Tbg*ISG75 SAXS data and analysis. The first panel represents the *Tbg*ISG75 scattering curve (experimental data in black, experimental error in gray). The inset shows the pair distribution function, in which the maximum dimension (D_max_) is the largest non-negative value that supports a smooth distribution function, an ab initio bead model of *Tbg*ISG75, and the SAXS-derived structural parameters (^a^radius of gyration calculated using Guinier approximation, ^b^radius of gyration calculated using indirect Fourier transformation, ^c^Porod volume calculated using Guinier approximation, ^d^Molecular mass estimation, which corresponds well to the ^e^expected molecular mass of 49.6 kDa according to the amino acid sequence without the predicted glycosylation site). The second panel shows a normalized Kratky plot, which suggests that the particle is folded and contains flexible linkers. For comparison, an intrinsically disordered standard (Tau protein) and a globular standard (bovine serum albumin) are also shown.

**Figure 2 pathogens-10-01050-f002:**
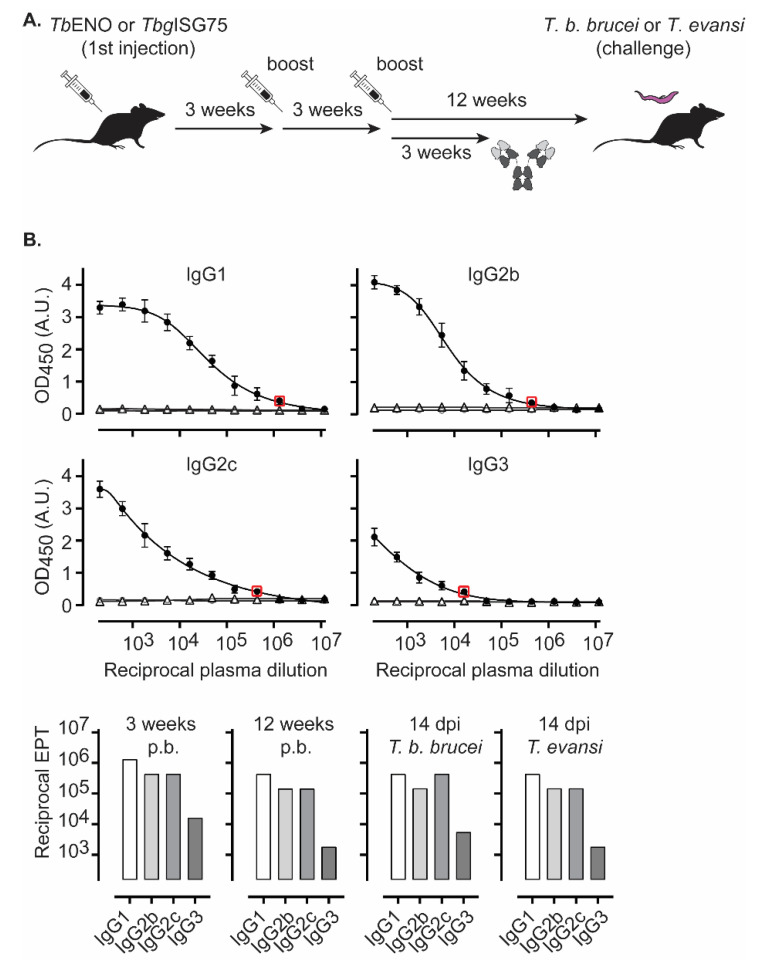
*Tb*ENO vaccinations induce high titers of circulating IgG1, IgG2b, IgG2c and IgG3 antibody titers. (**A**) Schematic overview of the employed vaccination schema. Plasma samples were collected 3 weeks after the third vaccine boost, and (**B**) specific anti-*Tb*ENO antibody titers were measured using antigen coated ELISA plates and 1/3 serial dilutions of plasma samples, starting at a concentration of 1/200 (closed circles: vaccinated, open circles: non-vaccinated, open triangles: mock vaccinated). Values are represented as the mean ± SD of four individual mice per timepoint. Endpoint titers were determined as the last dilution with an ELISA OD significantly higher than the background values (red box). Endpoint titers (EPT) that are visualized on the dilution curve plots of the samples obtained 3 weeks after the third boost are also represented in the bar-graph plot (bottom panel left). The same calculation method was used to determine endpoint titers 12 weeks after the third boost, just prior to the parasite challenge (bottom panel middle), and at 14 dpi, in both a *T. brucei* and *T. evansi* challenge setting (bottom panel right).

**Figure 3 pathogens-10-01050-f003:**
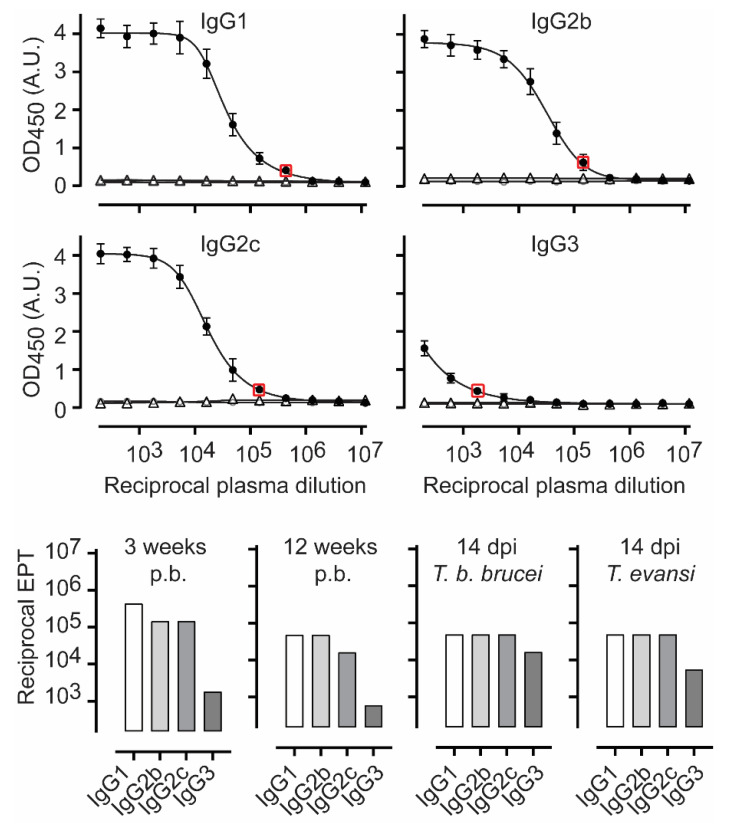
*Tbg*ISG75-vaccination induces high titers of circulating IgG1, IgG2b, IgG2c and IgG3 antibody titers. Plasma samples were collected 3 weeks after the third vaccine boost, and specific anti-*Tbg*ISG75 antibody titers were measured. Values are represented as the mean ± SD of four individual mice per timepoint (closed circles: vaccinated, open circles: non-vaccinated, open triangles: mock vaccinated). Endpoint titers (EPT) were determined as the last dilution with an ELISA OD significantly higher than the background values (red box). Endpoint titers are also represented in the bar-graph plot for the time points 3 weeks after the third boost (bottom panel left), 12 weeks after the third boost, just prior to the parasite challenge (bottom panel middle), and at 14 dpi in both a *T. brucei* and *T. evansi* infection setting (bottom panels right).

**Figure 4 pathogens-10-01050-f004:**
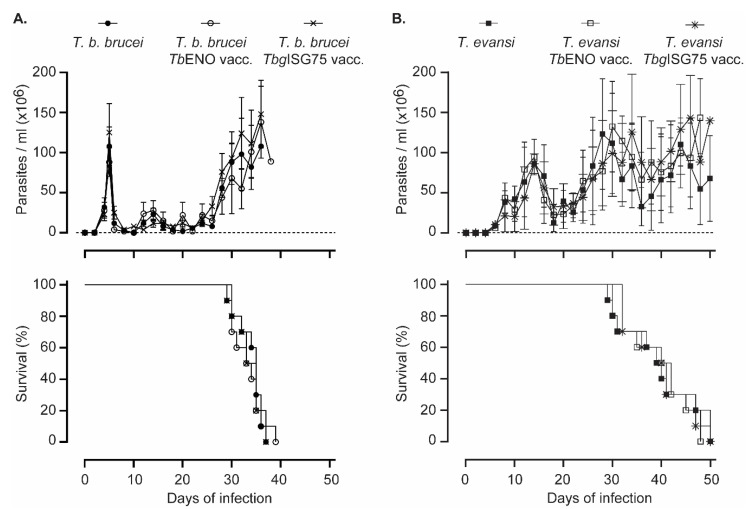
*Tb*ENO and *Tbg*ISG75 vaccination do not affect experimental trypanosomosis. Vaccinated and control mice were infected with trypanosomes through i.p. administration of 5000 *T. b. brucei* Anat1.1 parasites (**A**) or 200 *T. evansi* ‘Colombia’ parasites (**B**). Parasitemia was monitored at 2-day intervals and presented as mean ± SD of 10 mice and survival was monitored for all experimental groups, showing an overall lack of impact of either vaccination strategy.

**Figure 5 pathogens-10-01050-f005:**
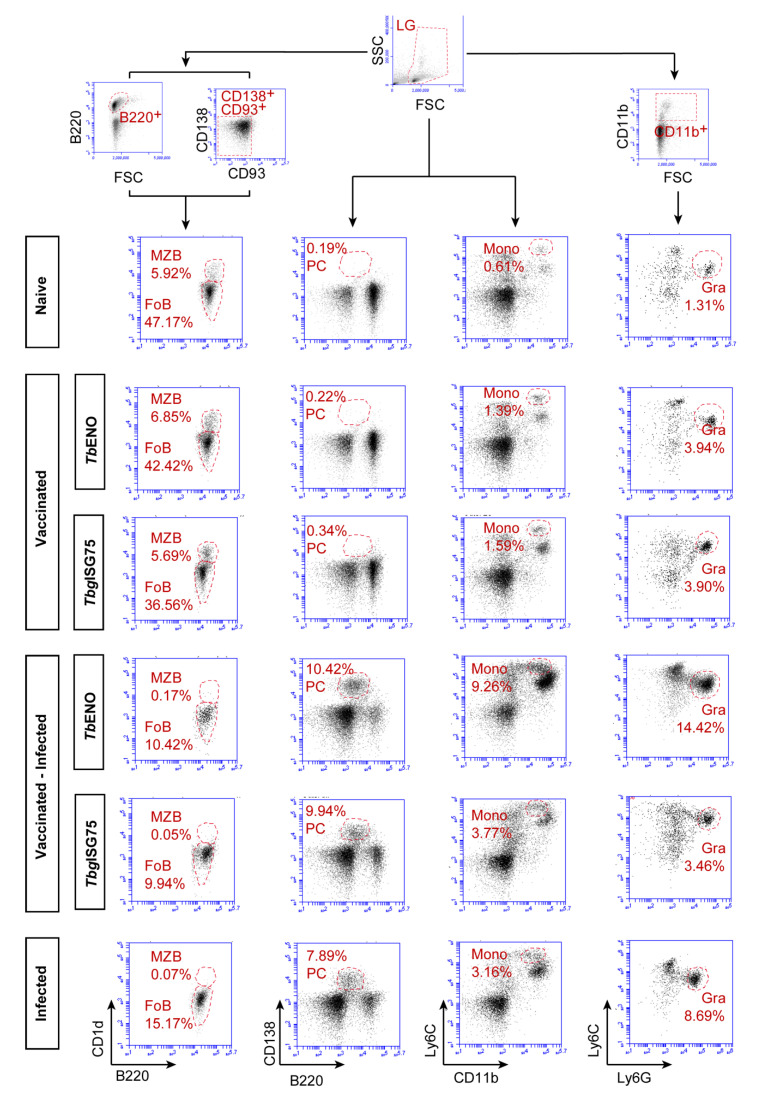
Flow cytometry analysis shows that neither *Tb*ENO nor *Tbg*ISG75 vaccination can prevent trypanosomosis-associated immunopathology. Spleens of naïve and vaccinated mice, as well as 14 dpi infected and vaccinated infected mice were analyzed using a set of B-cell markers as well as markers for monocytes and granulocytes. While the B220^+^CD1^+^CD93^-^CD138^-^MZB populations have been virtually obliterated at 14 dpi, the B220^+^CD1d^-^CD93^-^CD138^-^FoB population shows a marked size reduction upon infection, in both the non-vaccinated and the *Tb*ENO and *Tbg*ISG75 vaccinated groups (left panels). In contrast, B220^+^CD138^+^ PCs significantly increased in population size, both in vaccinated and non-vaccinated infected mice (middle-left panels). The same infection-induced increase in cell populations was found for the CD11b^+^Ly6C^High^ monocytes (Mono, middle-right panels) and Ly6C^+^Ly6G^High^ granulocytes (Gra, right panels).

## Data Availability

The data presented in this study are available upon request from the corresponding author.

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
