# Peer review of "The History of Anti-Trypanosome Vaccine Development Shows That Highly Immunogenic and Exposed Pathogen-Derived Antigens Are Not Necessarily Good Target Candidates: Enolase and ISG75 as Examples"

_pathogens, 2021, doi:10.3390/pathogens10081050_

Round 1

Reviewer 1 Report

1. Magez et al. presented a research about two previous vaccine candidates against trypanaosome that despite being highly immunogenic and conserved, showed poor efficacy against trypanosomes in mice challenged with the parasite. Another key point in this article was despite the production of specific antibodies in mouse models, these antibodies could not neutralize the parasites in mice experimentally infected with trypanosome thus, this parasite could have some immune evasion mechanism that could inhibit or prevent neutralization.

The paper has impact that will generate tremendous interest among researchers, physicians, and public health professionals. Moreover, several studies further explored vaccination as a strategy worthy of pursuit in efforts to control veterinary (and perhaps human) trypanosomiases. The idea that ‘vaccines are not possible for trypanosome infections” is a long-standing position held by many in the field. The rigorous work provided in this paper does provide any research direction on that notion.

2. Kindly avoid self-plagiarism in the methodology section. Please paraphrase effectively. It shows great similarity to your previous publication: Radwanska, M.; Nguyen, H.T.T.; Magez, S. African Trypanosomosis Obliterates DTPa Vaccine-Induced Functional Memory So That Post-Treatment Bordetella pertussis Challenge Fails to Trigger a Protective Recall Response. Vaccines 2021, 9, 603. https://doi.org/10.3390/vaccines9060603

3. Mice were injected with 5000 parasites, how to assure that this is the exact amount injected?

4. In a recent research, it is possible to elicit apparently sterile protection to an experimental trypanosome infection with a subunit vaccine that corresponds to the ectodomain of the invariant cell-surface parasite protein IFX. Kindly include this in your discussion part.

Reference: Autheman, D., Crosnier, C., Clare, S. et al. An invariant Trypanosoma vivax vaccine antigen induces protective immunity. Nature 595, 96–100 (2021). https://doi.org/10.1038/s41586-021-03597-x

Reviewer 2 Report

Trypanosomiasis represents a public health problem worldwide. It is urgent the identification of new drugs and vaccines as the only treatments available present limited efficacy, frequent treatment failures and side toxic effects, and no vaccine has been successful so far. Because of this, the current work is a valuable contribution to the difficult trajectory towards new vaccines for the neglected tropical diseases, being of relevance and general interest to the readers.

In overall, the manuscript is well written, the tittle is accurate, the abstract provide a precise summary, the assays are adequate and well conducted, and the structure is well defined. In summary, I consider this work to be of high quality. Hence, I recommend this manuscript for publication after minor revision. I explain my concerns in detail below.

Minor comments:

  1. Page 4, Lines 154-156. Why only 200 parasites were used for evansi infections? Was an intraperitoneal injection also performed? Please specify.
  2. Page 4, Lines 158-159. Why was parasitemia recorded up to 38 and 50 days post-infection for brucei and T. evansi, respectively? Was it because the parasitemia was never detected again? Several buffers allow quantification of parasitemia by making minor dilutions of the blood. Even a 1/100 dilution using just PBS is enough for a proper determination.
  3. Page 6, Lines 260-261. Please, indicate when mice were infected with evansi after vaccination.
  4. Page 10, Figure 2, legend. Please, define “dpi”.

Round 2

Reviewer 1 Report

The authors addressed all of my comments.

I recommend this research for publication.